# Climate Change, Health and Mosquito-Borne Diseases: Trends and Implications to the Pacific Region

**DOI:** 10.3390/ijerph16245114

**Published:** 2019-12-14

**Authors:** Walter Leal Filho, Svenja Scheday, Juliane Boenecke, Abhijit Gogoi, Anish Maharaj, Samuela Korovou

**Affiliations:** 1Research and Transfer Centre Sustainable Development and Climate Change Management, Hamburg University of Applied Sciences, Faculty of Life Sciences, 20, 21033 Hamburg, Germany; svenja.Scheday@haw-hamburg.de (S.S.); Juliane.Boenecke@haw-hamburg.de (J.B.); 2Department of Natural Sciences, Manchester Metropolitan University, Chester Street, Manchester M1 5GD, UK; 3Umanand Prasad School of Medicine and Health Sciences, The University of Fiji, Saweni, Lautoka 0700, Fiji; abhijitg@unifiji.ac.fj (A.G.); samuelak@unifiji.ac.fj (S.K.); 4School of Science and Technology, The University of Fiji, Saweni, Lautoka 0700, Fiji; anishm@unifiji.ac.fj

**Keywords:** climate change, global warming, human health, zika, pacific, extreme events

## Abstract

Climate change is known to affect Pacific Island nations in a variety of ways. One of them is by increasing the vulnerability of human health induced by various climate change impacts, which pose an additional burden to the already distressed health systems in the region. This paper explores the associations between climate change and human health on the one hand, and outlines some of the health care challenges posed by a changing climate on the other. In particular, it describes the links between climate variations and the emergence of climate-sensitive infectious diseases, such as the mosquito-borne diseases dengue, chikungunya, and Zika. The paper also presents a summary of the key findings of the research initiatives Climate Change and Prevalence Study of ZIKA Virus Diseases in Fiji and the findings from the World Mosquito Program as two examples of public health action in the Pacific region.

## 1. Global Climate Change and Its Toll on Pacific Island Nations

Climate change is a result of global warming, and is often defined as the change of temperature and weather patterns over a long period. In fact, the average global temperature in 2006–2015 was 0.87° Celsius higher than between 1850–1900 [1]. One of the significant causes of current climate change is human activities such as the burning of fossil fuels in industry and transport, deforestation, and land-use changes. Due to its scope, climate change is believed to be one of the most pressing global challenges of modern times.

In particular, the burning of fossil fuels increases the concentration of carbon dioxide (CO_2_) in the atmosphere, which is one of the gases that enhances the greenhouse effect. In 2017, 36.15 million tonnes of CO_2_ was emitted worldwide [2]. Further impacts of climate change are more frequent extreme weather events including flooding and droughts, erratic rainfall patterns, natural disasters, and melting glaciers contributing to sea level rises. According to the study “Ice Sheet Mass Balance Inter-comparison Exercise“, conducted by NASA and ESA, there is a recorded loss of 241.4 billion tons ice per year in Antarctica since 2012 [3]. Therefore, natural habitats, on the one hand, and living conditions of the human population on the other, are changing drastically, especially in coastal urban areas. Altogether, these effects will jeopardize the security of food, water availability, and air quality and may thus foster the emergence of climate-sensitive health disorders, including communicable and non-communicable diseases [4,5].

Pacific Island countries (PICs) are among the most vulnerable nations due to their exceptional exposure to climate change impacts and increasing weather extremes, with a substantial part of the population residing in highly susceptible coastal areas [6]. At the same time, resources to strengthen adaptation strategies, improve social and health care infrastructure, and to increase literacy among professionals and the population are lacking in order to cope with the rapidly rising burden of climate change [7]. For example, the absence of financial and material support, as well as insufficient knowledge transfer, hinder communities from developing proper adaptation measures to protect themselves from climate change impacts, leaving them unable to build adequate sea defenses or climate-resilient housing to withstand extreme events [8].

Among the major effects of global warming, mention can be made to the rapid increase in sea-levels, a matter of direct relevance to island nations, especially since glaciers are melting at a faster rate than ever before [9]. This poses a significant threat to inhabitants of PICs, where changes in water levels as little as 0.5 m as part of storm surges may lead to considerable coastal degradation and damages to properties and livelihoods. This is mainly attributed to inhabitants being located on the narrow fringes of the coastlines, where the majority of the economic activities occur as well. Those affected most badly, however, are the low-income sectors of the population. Additionally, the geological effects of erratic weather foster beach erosion, even further accelerated by the increase in commercial activities in beach areas cleared for sand extraction for buildings and areas of recreational activities [8].

Across the Pacific region, tropical storms, cyclones, and flash floods have become more frequent. This may cause damages to local infrastructure, result in increases of accidental deaths, escalate the chance of (re-)emerging epidemics, and leave many people homeless due to unprecedented flooding. On the other extreme, decreased rainfall has been recorded more often, resulting in prolonged droughts [10]. Communities relying solely on rain-water as their source of drinking water and farming may suffer most during drought-stricken periods. Besides, the occurrence of saltwater intrusion as a consequence of flooding threatens the quality of freshwater supplies and crop production in the Pacific islands [9,10]. Table 1 outlines some of the influences of climate change to Pacific States.

In addition, coral reefs are among the primary tourist sites in PICs, bringing in large sums of revenue per annum as one of the strongest local economic sectors. The effects of climate change, such as increasing sea temperature, sea-level rise, higher CO_2_ levels, and more frequent extreme events causing physical damage, compromising the health of coral reefs, leading to coral bleaching. The loss of these ecosystems and biodiversity puts health, well-being, and beauty on land and underwater environments at risk [11]. In addition to tourism, fisheries productivity is likely to be affected as coral reef ecosystems provide a quarter of the fish used for trade. Local fish population, such as tuna, are an important food and economic source in PICS that is affected by climate change. Increasing ocean acidification, fluctuations in ocean temperatures, and currents irregularities due to increases in El Niño–Southern Oscillation-like conditions may effect tuna population size or habitat distribution [10,12].

## 2. Climate Change and Its Challenges for Human Health and Well-Being

There are clear signs that climate change has adverse impacts on human health, with direct and indirect effects and both environmental (e.g., safe drinking water) as well as social determinants (e.g., poverty). These currently affect large numbers of people round the world. The World Health Organization (WHO) predicts that there will be approximately 250,000 additional deaths worldwide per year due to climate change between 2030 and 2050 [13], counting for 38,000 additional deaths due to heat exposure in elderly people, 48,000 additional deaths due to diarrhea, 60,000 additional deaths due to malaria, and 95,000 additional deaths due to childhood undernutrition. The increased burning of fossil fuels, as part of greenhouse gas emissions, may also result in ozone depletion leading to greater exposure to harmful ultra-violet radiation. This may cause conditions such as skin cancer, cataracts, and blindness [5].

The health and well-being of everyone can be affected due to climate change; however, especially children and elderly are considered at risk. Environmental health exposures, such as high temperatures or extreme weather events, amongst others, are putting large portions of the global population at risk of various negative health effects [14]. To illustrate this trend, two examples primarily exemplifying the adverse impacts and challenges of climate change on health are given:

*Heatwaves:* Periods of extreme increased temperature are occurring more often and affect health and well-being in various ways [15]. Exposure to extreme heat can result in dehydration, heat cramps, heatstroke, as well as chronic conditions such as cardiovascular and respiratory diseases. Moreover, it is expected that the prevalence of malnutrition will increase due to droughts and shifts in agriculture. Furthermore, increased temperatures are known to accelerate the activity of infectious agents and expand their geographical range while reducing human immunity, thus, increasing morbidity and mortality [5]. Some populations are more susceptible to the effects of temperature, including athletes and outdoor workers. Besides, children, older people, pregnant women, and those unable to control their body temperature due to pre-existing medical conditions, are among the most vulnerable to increased temperatures [7]. Studies have furthermore demonstrated that exposure to more extended periods of heat had more severe adverse health effects compared to instantaneous increases in temperature, explaining elevated mortality in urban areas due to heat accumulation and urban heatwaves than suburbs and rural areas [5]. Increasing temperatures also expand the abundance and season of allergenic pollens, thus, the exposure to allergic respiratory diseases, with aeroallergen levels, which can provoke asthma, becoming higher [13,16].

*Extreme weather events:* Due to extreme weather events such as tornadoes, flooding events, or cold waves, local infrastructures can be severely weakened, or even destroyed. As one result, sanitation and access to clean water can be affected, resulting in a lack of hygiene and the occurrence of communicable infections. Flash floods, for instance, can contaminate drinking water supplies by transporting bacteria, parasites, and viruses into clean water systems, thus, leading to outbreaks of waterborne diseases such as cholera, typhoid fever, or hepatitis A [13]. In addition, stress and trauma caused by extreme events may influence mental health and well-being. The WHO predicts that countries with a fragmented health infrastructure are among the least able to cope with climate change health impacts when no assistance to prepare and respond will be provided [13]. Extreme events, including heat and flooding, also promote the spread of vector-borne diseases, transmitted by pathogen-vectors such as mosquitos, ticks, and fleas [4,13]. Increased temperature increase vector and pathogen metabolism thus allow for faster replication and spread, whereas erratic rainfall patterns may increase the availability of suitable breeding sites [17,18]. Those trends may expand the transmission season in tropical and sub-tropical areas and create the risk of establishment of invasive vector species and infectious diseases in temperate climates by making the environment more suitable for the proliferation of the vectors [19] a trend that could also be observed in the tropical Pacific region.

## 3. Methodology

As earlier stated, the links between human health and climate change in Pacific island countries have been discussed in previous works (e.g., World Health Organisation 2015 [20]). In particular, the extent to which climate change may lead to a greater incidence of diseases, which can be considered “climate related”, was discussed in a recent study (Leal Filho et al. 2017 [19]). However, research on climate stressors as a whole, and on the surge of vector-borne diseases in particular, is not as abundant as it could be. Figure 1 illustrates some of the so called “climate stressors” and lists some of the groups which may be affected by them.

Based on the paucity of research specifically aiming at investigating diseases transmitted by vectors, more specifically mosquito-borne dieseases in the Pacific region in a climate change context, a study was performed on a sample of recent case studies and projects focusing on vector-transmitted diseases in the Pacific region, with an emphasis to mosquito-borne diseases. The reason for this focus is because such diseases are not as widely documented in the Pacific region as other communicable diseases are. The study consisted of three main elements:(a)an analysis of the literature of case reports and research projects specifically concerned with the connections between climate change and vector-borne diseases in the Pacific region;(b)the identification of examples of recent mosquito-borne outbreaks and public health activities in the Pacific region, as case studies;(c)the description of their nature, i.e., the main factors driving the outbreak, the projects’ focus, and their main challenges and implications for outbreak management and control in public health

References were identified through a narrative literature review using PubMed and Google Scholar searches limited to English articles only and published during the past 15 years. The keywords used included “Zika”, “Dengue”, “Chikungunya”, “Pacific”, “Pacific Islands”, “*Aedes* mosquito”, “Vector control”, “Vector”, and “Climate Change”. Additional scientific reports, e.g., published by IPCC, NASA, ECDC, or WHO, were obtained using Google Search in order to further include analytical and summary reports provided through specialized national and international agencies. Again, the search was limited to English reports and materials published during the past 15 years. A total of 42 articles and reports resulting from the searches were reviewed, and the ones deployed for this paper are quoted in it. The description of a set of case studies, with a focus on outbreaks of mosquito-borne diseases in Pacific Island countries, was complemented by an overall analysis of the results, and the identification of some of the common features of public health initiatives in the field of climate change and mosquito-borne diseases. The initiatives chosen include the research projects Climate Change and *Prevalence Study of ZIKA Virus Diseases in Fiji*, and the World Mosquito Program, both conducted in the Pacific region.

## 4. Results and Discussion

A total of 42 articles and reports resulting from the searches (key words: Climate change and health in the Pacific, mosquito-borne disease in the Pacific) were reviewed. Apart from the obvious link to the Pacific region, papers were selected under the criterium of thematic relevance, since papers dealing with mosquito-borne diseases in other geographical regions were not considered. The papers deemed adequate and suitable to be deployed for this paper, are referred to in the text.

First and foremost, the analysis of the literature has shown that, among the many vector-borne diseases in the Pacific, including those transmitted by mosquitoes, mention can be made to; dengue fever, chikungunya fever, Zika, Lymphatic Filariasis, Japanese encephalitis, Murray Valley encephalitis, Barmah forest virus disease, Ross River fever, and malaria [21]. In a global context, these diseases are more widely spread in tropical regions, where the combination of a warm and humid climate provide ideal conditions for the various types of disease carrying mosquito species to breed and survive, including *Aedes*, *Culex,* and *Anopheles* mosquitoes. In particular, in the Pacific region, the *Aedes* mosquito is a well-established vector of several viruses, primarily Zika (ZIKV), dengue (DENV) and chikungunya virus (CHIKV).

Development and life cycles of mosquitoes, and the pathogens they carry, are highly dependent on local ecosystems. *Aedes* mosquitoes are cold-blooded insects that require specific temperature ranges to survive, as well as water reservoirs to breed and develop [22]. Tropical and subtropical climate conditions characterized by elevated temperature, prolonged rainfall, and high air humidity hence serve as an exceptionally suitable environment for mosquitoes’ survival, development, and replication [23]. For example, erratic rainfall patterns can positively or negatively influence the emergence of mosquito-borne outbreaks. Increased rainfall and flooding events may increase the potential vector breeding sites and habitats for mosquito larvae. However, excessive rainfall may result in flash floods, which may destroy mosquito eggs or wash away larvae [17]. As virus carrying mosquitoes require aquatic breeding sites, a scarcity in rainfall and prolonged droughts may lead to desiccation of larvae [24]. Especially during dry seasons, populations in tropical countries tend to store rainwater, which forms a stagnant water environment suitable for mosquitoes to lay eggs, hence maintaining potential breeding sites even in dry conditions. The development of mosquito larvae is furthermore promoted with increased surrounding temperature, allowing for faster maturation. Moreover, adult female mosquitos feed more often on blood and thus may transmit infectious agents with higher frequency [25]. Viral replication within the mosquitos is furthermore increased, leading to a reduced extrinsic incubation time with a rise in environmental temperature. As a result, warmer temperature conditions mainly during the rainy season, both contributing to high humidity levels, allow tropical mosquitos to flourish, thus, promoting the occurrence of *Aedes*-borne outbreaks and persistent virus circulation between host and vector populations [26,27].

### 4.1. Mosquito-Borne Diseases in the Pacific Region—A Review on DENV, ZIKV, and CHIKV

In the Pacific region, *Aedes* mosquitoes have been identified as the main disease vector species. Mainly causing dengue epidemics linked to a single DENV serotype throughout the past, recent research identified the co-circulation of not only different dengue virus serotypes (types one to four), but newly emerging *Aedes*-borne Zika and chikungunya transmission [28]. Regular DENV outbreaks had been reported since 1960s from the Pacific region. Since then, DENV has established endemic and epidemic circulation. In the early 21st century, ZIKV (since 2007) and CHIKV (since 2011) epidemics occurred for the first time [21]. The most important mosquito-borne viral genera in the Pacific region include *Alphaviruses* (family *Togaviridae*), e.g., chikungunya virus, and *Flaviviruses* (family *Flaviviridae*), e.g., Zika and dengue virus. Those viruses could primarily be linked to the *Aedes aegypti* mosquito, which is mainly present in human residential areas, and *Aedes albopictus*, mainly found in peri-urban or rural areas [29].

DEN, ZIKV, and CHIKV are tropical mosquito-borne viruses that share biological similarities with other viral agents such as Ross River virus (closely related to CHIKV), West Nile virus, and yellow fever virus (closely related to DENV and ZIKV). They primarily spread through the bite of disease-carrying female mosquitoes of the genus *Aedes* (Stegomyia), such as *Aa. aegypti*, *Ae. africanus*, *Ae. hensilli*, and *Ae. albopictus* [30]. Unlike other flaviviruses, ZIKV is known to further be passed from human to human through perinatal, congenital, sexual, and transfusion transmission [31]. Blood-borne transmission and intrapartum transmission have also been reported for CHIKV [32]. The main reservoir hosts include human and non-human primates and may further be present in other wild and domestic animals. The incidence and severity of mosquito-borne arbovirus infections that can be maintained in human-vector-human-cycles (with no involvement of animals in the transmission cycle) such as DENV, ZIKV, and CHIKV, has increased in the Pacific regions during the past years [28]. Especially in populations with poor herd immunity (high susceptibility due to low seroprevalence levels in the population) and lacking resilience capacities, the (re-)emergence of mosquito-borne diseases (MBD) is particularly hazardous to human health, often leading to major outbreaks. 

As far as the identification of outbreaks is concerned, a further aim of this study, Table 2 describes the nature of some recent DENV, ZIKV, and CHIKV outbreaks in different Pacific Island States, with an emphasis on the associated public health challenges and implications.

Despite their epidemic potential, the clinical picture of DENV, ZIKV, and CHIKV is often mild, and symptoms often go unrecognized [37]. Mild presentation of all three viral infections includes flu-like symptoms, with fever, rash, and malaise, which often results in misdiagnosis when laboratory confirmation is missing [37,38]. Most people recover fully, however, in rare cases infection may cause severe health complications such as encephalitis, neurological disorders, infantile malformations (ZIKV), severe and long-lasting arthralgia (CHIKV), or hemorrhagic manifestations (DENV), leading to hospitalization or fatal complications. Yet, the interaction between the different pathogens is poorly understood. At present, for all three mosquito-borne viral diseases there is no antiviral treatment [39], however, there are advances for the control of these diseases, e.g., the first licensed dengue vaccine and other vaccine candidates [40]. Personal protection, such as wearing long-sleeved shirts and pants, application of insect repellents, insecticides spraying, bed nets application, as well as consequent cleaning of mosquito breeding sites remain the primarily prevention strategies. However, to implement such strategies successfully, the continuous mobilization of affected communities is required [41].

With increasing climate change, climate-sensitive infectious diseases, particularly MBD, are on the rise. In the case of the Pacific region, the incidence and severity of MBD outbreaks have significantly increased during the past years [28], calling for a better understanding of how climate change is shaping the future burden of mosquito-borne diseases in highly susceptible areas such as Pacific island nations. The 2016 ZIKV epidemic in the Americas fueled the debate how climate change may impact the range of disease vectors and how transmission seasons will be affected, especially in highly suitable tropical environments. Against this backdrop, the research initiative *Climate Change and Prevalence Study of ZIKA Virus Diseases in Fiji*, which has been conducted by the authors and was funded by the German Federal Ministry of Education and Research (BMBF), aimed at investigating the outbreak of emerging ZIKV disease in the Fiji Islands from an ecological perspective (https://www.haw-hamburg.de/en/ftz-nk/projects/zika-fiji-en.html). Table 3 describes the main implications of the research initiative based on the preliminary results.

### 4.2. Vector and Transmission Control in the Pacific Region—The Wolbachia Method

According to recent reports, 2019 has been an exceptionally active year of DENV outbreaks worldwide [45]. As outlined, with increasing climate change and international mobility, such trends may become more likely, with prolonged transmission seasons, whereas invasive vector species and newly emerging viruses may infest and establish in yet non-affected but increasingly suitable areas [46]. Addressing the health challenges of mosquito-borne epidemics is hence one important element in the quest for greater climate resilience [47].

As most prevention strategies are mainly based on personal protective behavior, often requiring behavioral modification and significant awareness among the population, novel entomological approaches to cut off transmission routes for DENV, ZIKV, and CHIKV disease gain increasing attention in public health. Control of disease-carrying mosquito populations through the blockage of virus replication within *Aedes* mosquitoes showed a significant reduction in the level of virus transmission by *Aedes aegypti* [48]. Indicating promising results in field trials and laboratory research, scientists are aiming to minimize the risk of (re-)emerging epidemics in pilot projects throughout the Pacific region. Through the infection of *Aedes aegypti* using *Wolbachia* bacteria strains, the vector competence for viruses, including ZIKV, DENV, and CHIKV, can be strongly reduced by inhibiting virus replication inside the mosquitoes, even under highly conducive environmental conditions [49]. Considering the climate-sensitivity of mosquito vectors and mosquito-borne viruses, such programs may benefit from knowledge on local environmental suitability levels and transmission thresholds, e.g., by targeting local transmission pockets during high seasons to release *Wolbachia* infected mosquitoes and ensure effective establishment. Recent achievements of the World Mosquito Program (http://www.eliminatedengue.com/pi), which is also working in the Pacific region implementing the *Wolbachia* method, are described in Table 4.

The data and evidences gathered point to three main trends:

Firstly, climate is a critical factor in the spread of vector-borne diseases, especially if it is taken into account that it increases the risks and facilitates the expansion of MBD epidemics. The sustained changes being observed in the global and regional climates, as well as the non-climatic factors such as fragile health systems and insufficient surveillance systems, indicate that the Pacific region is exceptionally vulnerable to the (re-)emergence of MBDs.

Secondly, the Pacific region is characterized by a significant suitability to persisting mosquito-borne virus (co-)circulation, which can be attributed to some elements such as:(a)highly conducive climatic conditions, especially temperature increases combined with sustained periods of rainfall, which are factors that facilitate the reproduction and spread of mosquitoes as vectors, whose biology and the replication of the viruses they host are highly temperature- and moisture-dependent;(b)inadequate living and sanitation facilities, particularly susceptible to disruption and destruction during natural disasters, which can be found in many countries in the region, offering suitable breeding conditions for the spread of virus-carrying mosquitoes and other infectious diseases.

Thirdly, whereas the associations between climate variability, climate change, and the transmission of MBDs are fairly well studied, much work is still needed in incorporating research findings into health prevention policies. in view of future climate change adaptation scenarios, which are important to Pacific nations [51].

## 5. Conclusions

This paper has illustrated how climate change affects Pacific nations and the main links between climate change and health. As a particular issue, it also described the examples provided by mosquito-borne diseases, especially dengue, chikungunya, and Zika. The key findings of the project *Climate Change and Prevalence Study of ZIKA Virus Diseases in Fiji* were also presented. It is seen that climate change poses various threats to many island nations in the Pacific region, one of which being the spread of mosquito-borne diseases as a whole. It can be concluded that, apart from deeper insights into how climate change may influence the spread of vectors, a better understanding of the various impacts of climate change to human health is needed, especially in areas where sentinel health surveillance and early warning systems are lacking, which may guide suitable adaptation measures. These are considered essential in order to better prepare Pacific nations to cope with the many challenges climate change poses to them.

## Figures and Tables

**Figure 1 ijerph-16-05114-f001:**
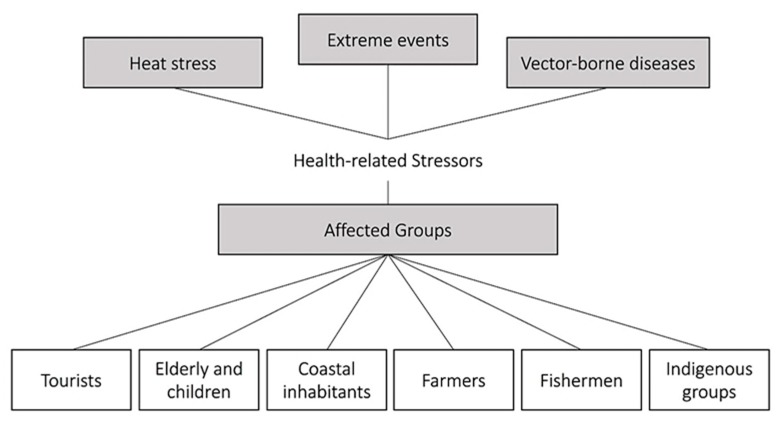
Some of the climate related stressors and affected groups in the Pacific region.

**Table 1 ijerph-16-05114-t001:** Some of the influences of climate change to Pacific islands.

Sector	Examples of Influences
Agriculture	Variations in rainfall and crop production, often reduction of yields
Tourism	Extreme events may reduce numbers of visitors
Transport	Extreme events impair road, sea, or air transport
Real Estate	Extreme events may cause damages to property and reduce their value
Social services	Climate distresses may lead to inland migration and social unrest
Health	Spread of vector- and water-borne diseases, mental problems associated with direct injuries and private losses (e.g., damages to properties), and extreme heat events associated with non-communicable diseases such as circulatory diseases, including increased pulmonary effects of heat and air pollution
Economics	Increased hardship as a result of financial losses due to extreme events

**Table 2 ijerph-16-05114-t002:** Case studies of recent Aedes-borne outbreaks in Pacific Island States.

Pathogen.	ZIKV [21,29,33]	DENV (DENV-2) [21,29,34]	CHIKV [35,36]
Pacific Case Study	Yap Island, Federate States of Micronesia, 2007	Solomon Islands 2016/2017	French Polynesia 2014/2015
Population	Approx. 7.400 (2000 census data)	Approx. 640.000 (2016 projection)	Approx. 276.000 (2017 census data)
Main vector	*Ae. hensilli*	*Ae. aegypti*	*Ae. aegypti, Ae. polynesiensis*
First reported in the Pacific	2007, Yap Island; major outbreaks 2007-2017	1912, Hawaii; regular outbreaks since 1960s	2011 New Caledonia; major outbreaks 2011-2015
Outbreak period	Apr 2007–Jul 2007	Aug 2016–Apr 2017	Sep 2014–Mar 2015
Extent of the outbreak	49 confirmed cases >900 suspected cases	1,510 confirmed cases, 12,329 suspected cases,	4,443 confirmed cases, 69,000 suspected cases
Public Health Challenges	Widespread mosquito vector; immunologically naïve population; co-circulating DENV	Widespread mosquito vector; endemic dengue virus (DENV) circulation; outbreak quickly consumed public and clinical resources	Widespread mosquito vector; immunologically naive population; co-circulating DENV
Mosquito-borne virus co-circulation	DENV, zika virus (ZIKV)	ZIKV	DENV, ZIKV, Ross River virus (silent)
Public Health Implications	Development of robust surveillance systems (health and vector surveillance); increase of diagnostic capacities and training; implementation of vector control measures, including individual and collective protection; and increased awareness and community engagement concerning mosquito-borne diseases		

**Table 3 ijerph-16-05114-t003:** Key findings of the research project “Climate Change and Prevalence Study of ZIKA Virus Diseases in Fiji”.

Aim: Research shows that the global temperature rise, changes in precipitation patterns and increasing global trade and travel facilitate range expansion of MBD, and may further extend transmission seasons in endemic areas, like the Pacific region. As a consequence, emerging diseases, such as ZIKV disease, have successfully expanded to geographical areas where only DENV epidemics used to occur, including the Fiji Islands. The project aimed at exploring ZIKV environmental suitability and potential of (re-)emergence in the Fiji Islands looking at population susceptibility, variations in temperature, and rainfall levels on the main island Viti Levu.				
Methods: Meteorological data from 11 weather stations (Ø minimum and maximum temperature °C, total precipitation mm/km2, 1960–2018) and reports on confirmed DENV and ZIKV infection (2007–2018) were acquired on a monthly basis in order to describe overall meteorological and epidemiological trends in Viti Levu, Fiji. Additionally, evidence from recent ZIKV serological studies was acquired to explore overall ZIKV susceptibility levels in the Fiji population. Environmental DENV and ZIKV epidemic exposure was then assessed using pathogen-specific thermal thresholds derived from laboratory studies (ZIKV = 22.6–34.8 °C, DENV = 17.8–34.6 °C) [41] looking at meteorological trends 1960–2018.				
Key findings: Recent findings could only confirm low-level transmission of ZIKV in Fiji during 2013–2017, indicating low-level herd immunity in the Fiji population [42,43]. Overall transmission season could be observed from November to June (warm and wet season), with highest counts of DENV and ZIKV infection recorded in Fiji’s densely populated areas, mainly after rainfall events. Moreover, meteorological records (1960–2018: Ø Tmin =20.8 [+0.3] °C; Ø Tmax = 29.1 [+0.3] °C) indicate suitable temperature conditions for DENV year-round transmission since 1960, especially in Fiji’s main urban areas in the South-East and North-West of Viti Levu, whereas only seasonal ZIKV transmission risk could be identified, especially due to minimum temperature levels dropping beyond the ZIKV transmission threshold levels.				
Pathogen	Time	Tmin suitability (1960–2018)	Tmean suitability (1960–2018)	Tmax suitability (1960–2018)
DENV (17.8–34.6 °C)	No months/year	11.2 (–0.4)	12.0 (+/–0)	11.8 (+0.2)
ZIKV (22.6–34.8 °C)	No months/year	1.9 (+2.1)	10.6 (+0.4)	11.5 (+0.3)
With endemic ZIKV circulation in Southeast Asia [44], and as temperature rises, there is a risk of ZIKV re-emergence in Fiji, with potentially extended transmission season in the near future. The findings demonstrate that ZIKV must still be considered a potential health threat in the Pacific, although no infected cases have been reported recently, which makes awareness raising and epidemic preparedness a public health priority.				

**Table 4 ijerph-16-05114-t004:** The Wolbachia approach in the Pacific region [50].

Aim: The *World Mosquito Program*, with a focus on the Pacific region, aims to eradicate viruses like DENV, CHIKV, and ZIKV by cutting of the transmission routes for mosquito-borne diseases. By infecting larvae with a particular strain of the bacterium *Wolbachia* (wMel), the wild mosquito population as well as its vector capacities will decrease over time, as well as facilitate mosquito suppression, resulting in reduced virus transmission levels.
Methods: The *Wolbachia* approach is currently applied by the *World Mosquito Program* in Fiji, Vanuatu, Kiribati, and Sri Lanka. Mosquito larvae are infected in laboratories with a specific strain of *Wolbachia*, which is originally taken from *Aedes reversi*, a related *Aedes* species. After the infection, the *Wolbachia* carrying male mosquitoes are released in target areas in order to breed with wild mosquitoes and grow in population.
Results: *Wolbachia* transmission could be identified from infected male mosquitoes to wild female mosquitoes (horizontal), as well as from parent to offspring inside the mosquito eggs (vertical). As a result, two effects could be observed: First, *Wolbachia* strains inside the mosquito reduce the replication of viruses, such as DENV, ZIKV, and CHIKV. Second, mosquito eggs may not hatch and, consequently, mosquito populations may be suppressed. This method has been successfully applied in Guangzhou, China, where scientists nearly eliminated *Aedes albopictus* mosquitoes from two islands by using the *Wolbachia* technique, with promising long-term effects: Studies in inner Cairns suburbs already showed that the *Wolbachia* strain can exist in mosquito populations for more than eight years [49]. According to Hervé Bossin, the project’s lead scientist, the *Wolbachia* approach is expected to help solving the mosquito problem in Island States within the next ten years.

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
