# Peer review of "Climate Change, Health and Mosquito-Borne Diseases: Trends and Implications to the Pacific Region"

_ijerph, 2019, doi:10.3390/ijerph16245114_

Round 1
Reviewer 1 Report
Why convinience sampling strategy is adopted? how do you reach the sample size of 699? There is hardly any analyses of the data? Data is also not presented in descriptive form? which methods are used for analyses of data?Author Response
please see the attachment

Reviewer 2 Report
Dear Authors,
Thank you very much for the opportunity to read your manuscript describing the key findings of your project “Climate change and Prevalence Study of ZIKA virus diseases in Fiji”. I very much liked the manuscript, and it’s very well written. However, I was somewhat surprised by the overall structure and the brevity of the description of the actual study. The structure of the manuscript reminded me more of a project report than a peer-reviewed journal article manuscript, as it was lacking formal sections of Introduction, Methods, Results and Discussion. Most of the manuscript was providing an Introduction, while both Methods, Results and Discussion were contained in Box 1, which is unusual for a peer-reviewed manuscript. Was this perhaps adapted from a project report? Is there perhaps a manuscript developed that describes this study in more detail? In particular, I have a number of questions about the details of your study which were not addressed in Box 1, in addition to a few minor comments. Please see below:
On page 1, in the very first sentence, you use a reference for climate change that is from National Geographic, which is not a peer-reviewed publication. I’m sure you could find a better, peer-reviewed reference, perhaps from the IPCC reports, for the definition of climate change. Please do so! On page 1, first sentence of the second paragraph, you provide the chemical formula for carbon dioxide (CO2). Please use the subscript for the number 2, as CO2 On page 3, in the third paragraph, you talk about different mosquito vectors. Please italicize their name, and do so throughout the manuscript. On page 3, in the third paragraph, you have a sentence that says: “In particular, the Aedes aegypti mosquito, can transmit Zika, dengue, chikungunya, and yellow fever,..”. Please put the word “viruses” after the phrase “yellow fever”, to clarify what these pathogens are. On page 4, in the last paragraph, you state that “ZIKV can remain in semen for an extended period than any other body fluid…”. Please change this to “ZIKV can remain in semen for a more extended period than any other body fluid…”. In Box 1, I have a lot of questions. What was the brakedown of serological samples between different parts of Fiji? What were the exact locations? What were the characteristics of the participants? In particular, how do those compare to the overall population of these locations, especially given that you conducted convenience sampling? Why did you do convenience sampling, instead of a stratified design? Is there a cross-reactivity of the kit that you used with other flaviviruses such as dengue virus? Chikungunya is an alphavirus so there shouldn’t be cross-reactivity, please mention that. In Box 1, in the Key Findings section, you state that “further analysis (Fig 1 A/B) revealed higher numbers of infected cases especially in densely populated areas and after rainfall events”. Was this a statistically significant result? What statistical methods did you use? Neither of those are apparent on Figure 1. I understand that the lack of large numbers of cases can be an impediment to robust statistical analysis. Figure 1 is a bit too complicated, with the two maps separated by the legend of subfigure [c]. The same goes for the two subplots [b] separated by the subplots [c]. Please reorganize that different parts of the subplots are next to each other. In addition, I cannot see the lines for the dengue virus temperature range on subplots [c] of Figure 1. What is the green dashed line on subplot [c], is that the same as ZIKV Tmin? The lack of a discussion on these findings is disturbing. In particular, I would like to see a discussion on why seroprevalence rates and case numbers are so low for ZIKV and CHIKV in Fiji, when dengue viruses are endemic. Is there a potential cross-immunity against ZIKV or CHIKV in people who have already been infected with DENV? Is there an interaction of these viruses in the mosquito vectors? Is there a genetic component of protection in the local population? In the References, on the first page (labelled as 2 of 11), on line 27, for the reference for Asad & Carpenter, please capitalize “zika virus”.
Round 2
Reviewer 1 Report
Authors have addressed most of the comments provided by reviewer. I suggest accepting the paper.
Author Response
Thank You
Reviewer 2 Report
Dear Authors,
Thank you very much for submitting your revised manuscript. You obviously worked a lot on this revision, which I very much appreciate.
The biggest issue I currently have with your paper is that it is very difficult for me to decide what it is about. Is this a review of the literature on the connections between climate change and mosquito-borne diseases in the Pacific Region? Is this an opinion piece on the same? Or is it a lot of extra writing in order to introduce your study on ZIKA virus in Fiji? I wish you would have chosen one of those, as at this point it is a mixture of all of those, and therefore doesn’t do any of those proper justice! Please choose one of the above and stick to that. That would really help the quality of the manuscript, and would make it so much easier to review.
I do have a number of minor comments, please see them below:
First page, line 23, please capitalize “Zika”. On the second page, in Table 1, for the influences of climate change to Pacific Islands, in the Health Sector, I wonder why you don’t mention heatwaves and air quality, which you do mention in the text later on. On page 3, line 93, I wonder whether you really need the comma after “both”. I would suggest to remove it. On page 4, line 135, you use the term “pathogen performances”. I wonder what you mean by that? Do you mean “fitness”? Or “metabolism”? The term is very ambiguous as is. On page 4, line 136, you state that “erratic rainfall patterns increase the availability of suitable breeding sites”. As you write later in the text, that might not always be the case. Therefore, I would suggest to temper this statement by saying instead “erratic rainfall patterns MAY increase..”. On page 4, line 138, you use the term “carrier species”. Do you mean “vector species”? The term “carrier” is very ambiguous, as it might mean an infected host, a chronic infection, or a vector. I do think that you mean “vector”. If that is correct, please replace that, as well as throughout the text. On page 5, lines 154-155, you state that there is a “a paucity of research specifically aiming at investigating diseases transmitted by vectors in a climate change context”. Do you mean overall? Or specifically in the Pacific region? A search on Google Scholar with the keywords “vector-borne disease” “climate change” and “Pacific” results in 3310 results, but I have not investigated them, so I don’t know how many are relevant. So is there really a paucity? On page 5, line 158, you state that “such diseases are not as widely documented in the Pacific region as other communicable diseases are, to which they are associated with”. What do you mean by “to which they are associated with”? What is associated with what? The other communicable diseases to vector-borne diseases? Please clarify! On page 5, on lines 160-175, you discuss the methodology of your study as “references were identified through PubMed and Google Scholar”. So is this a review paper? Is this a systematic review? Or a meta-analysis? Please identify your methodology! On page 5, line 171, you state that “Additional scientific reports…were obtained using Google Search.” In this process, did you use the same keywords as for your search on PubMed and Google Scholar? What was the range in the terms of publication years that you focused on? In terms of the initiatives chosen to include, why did you include the two that you discussed. On page 5, Section 4, you talk about the results of your study. However, you don’t mention any of the usual results of a literature review or systematic review. For example, you don’t mention how many papers and studies you found in your search, and the criteria you used to select the one from those that you read and included. Or how many you found using different keywords, such as the different mosquito-borne diseases. Please remedy that. On page 5, line 179, I suggest to remove the first sentence of the Results and Discussion as it does not provide any information. On page 5, lines 181-183, you provide a list of mosquito-borne diseases in the Pacific. I was surprised not to see Zika virus on that list. Why is that? On page 5, line 200, you talk about “breeding choices”. Do you mean “breeding sites”? I would suggest to use that instead. On page 6, line 202, you state that “During breeding times, adult female mosquitoes feed more often on blood…”. Female mosquitoes only take blood during the gonotrophic cycle when they are developing eggs. Do you mean that in higher temperatures they feed more often on blood? That would be correct. On page 6, line 210, I would suggest changing “carrier” to “vector” again. On page 6, line 213, please capitalize “Zika”. On page 6, line 218, please capitalize “Zika” and fix “Flavirviruses” to “Flaviviruses” (so no “r” before second “v”). On page 6, line 222, please fix “West Nil virus” to “West Nile virus”. On page 7, line 243, you state that “the human-pathogenic potential of DENV, ZIKV and CHIKV is considered low for the majority of cases”. What do you mean by “low”? This is a very subjective statement. “low” compared to what? If it’s that low, why do we spend so much money on it? On page 7, line 250-251, you state that “At present, for all three mosquito-borne viral diseases there is no approved vaccine or antiviral treatment”. This is incorrect. The Dengvaxia vaccine has been approved for use in 11 countries as of 2016, as well as in the United States in 2019. On page 7, lines 256-265, you discuss the project “Climate Change and Prevalence Study of Zika virus diseases in Fiji”, which is the authors study. I would like you to make that explicit, as it is not clear at the moment. In addition, there is no reference to a published study, or even to a webpage of the project. Please provide that. On page 8, in the last sentence of Box 2, you use the phrase “awareness rising”. Do you mean “awareness raising” On page 9, line 292, please capitalize “box 2”. On page 9, in Box 2, in the expected outcomes, you describe some of the results of the World Mosquito Program using Wolbachia. However, these results have already been demonstrated. Why do you call these expected? You should just say that there are the results. On page 9, in Box 2, in the “Expected outputs” portion, you describe how Wolbachia reduced population decrease. However, Wolbachia also makes mosquitoes refractory to disease transmission. Please mention that as well. On page 9, line 296, you state that the “The data and evidences gathered point to three main trends:” Is this portion part of the Conclusions? I would suggest starting the Conclusions section here then. On page 9, lines 312-314 sounded confusing to me. You say that “the associations between climate variability, climate change and the transmission of MBDs are fairly well studied”, but at the same time “much research is still needed on the tools which may be deployed to quantify such associations”. How can we have the “associations…fairly well studied”, if we still need “much research…on the tools” which we use to “quantify such associations”? This does not make sense, please clarify! In the references, whenever you had a hyphenated word, my printer would print an empty box. Please fix those.Author Response
please see the attachment

Round 3
Reviewer 2 Report
There are a number of smaller issues that I'd like the authors to address in any case: 1. On page 2, line 44, you write that "living conditions of the human population...will change drastically". Are these living conditions already changing, or not yet? If the former, please indicate. 2. On page 3, lines 80-82, you write "Seeing the effects of climate change, increasing sea temperature, sea-level rise, higher CO2 levels, and more frequent extreme events causing physical damages are compromising the health of coral reefs, leading to coral bleaching." This sentence is hard to understand. I suggest instead: The effects of climate change, such as increasing sea temperature, sea-level rise, higher CO2 levels, and more frequent extreme events causing physical damage compromise the health of coral reefs, leading to coral bleaching. 4. On page 3, lines 85-89, is also a very long sentence that is difficult to understand. I suggest reversing the sentence structure, starting the sentence with "Local fish populations, such as tuna, ..." and then explain why. 5. On page 5, lines 173-174, you state that "A total of 42 articles and reports resulting from the searches were reviewed, and the ones deployed for this paper are herewith referred to." What do you mean by referred to? with a certain name? I don't understand. 6. On page 6, line 216, you state that "promoting the occurrence of Aedes-borne outbreaks and persistent circulation between host and vector populations". I suggest inserting "virus" in between "persistent" and "circulation", so we get "persistent virus circulation". 7. On page 7, in Box 1, in the left hand column, you discuss the 2007 ZIKV outbreak on Yap island, and state that ZIKV co-circulated with DENV and CHIKV. However, on the right hand column, you describe CHIKV, and you indicate that the first reported outbreak in the Pacific was in 2011 in New Caledonia. But if that's true, how could CHIKV co-circulate with ZIKV in Yap island in 2007? Or did it only co-circulate after 2011? Please clarify. 8. On page 7, lines 272-275, you describe your study. I suggest putting the term "is" before the word "is" to provide a verb. 9. On page 9, line 317, you say that "viruses they host". If you're talking about mosquitoes, those do not host viruses, they vector them. Please say "viruses they vector" instead. 10. On page 10, line 324-325, you state that " much research is still needed on the tools which may be deployed to quantify such associations" This is contrary to your cover letter, in which you state that for this section you replaced this earlier text with "much work is still needed in incorporating research findings into health prevention policies". Please correct this so that it is the same phrasing as in the cover letter.Author Response
see the attachment
